# Testing and Learning on Distributions with Symmetric Noise Invariance

**Ho Chung Leon Law**
Department of Statistics
University Of Oxford
hlaw@stats.ox.ac.uk

**Christopher Yau**
Centre for Computational Biology
University of Birmingham
c.yau@bham.ac.uk

**Dino Sejdinovic**
Department of Statistics
University Of Oxford
dino.sejdinovic@stats.ox.ac.uk

## Abstract

Kernel embeddings of distributions and the Maximum Mean Discrepancy (MMD), the resulting distance between distributions, are useful tools for fully nonparametric two-sample testing and learning on distributions. However, it is rare that all possible differences between samples are of interest – discovered differences can be due to different types of measurement noise, data collection artefacts or other irrelevant sources of variability. We propose distances between distributions which encode invariance to additive symmetric noise, aimed at testing whether the assumed true underlying processes differ. Moreover, we construct invariant features of distributions, leading to learning algorithms robust to the impairment of the input distributions with symmetric additive noise.

## 1 Introduction

There are many sources of variability in data, and not all of them are pertinent to the questions that a data analyst may be interested in. Consider, for example, a nonparametric two-sample testing problem, which has recently been attracting significant research interest, especially in the context of kernel embeddings of distributions [2, 5, 7]. We observe samples $\{X_{1j}\}_{j=1}^{N_1}$ and $\{X_{2j}\}_{j=1}^{N_2}$ from two data generating processes $P_1$ and $P_2$, respectively, and would like to test the null hypothesis that $P_1 = P_2$ without making any parametric assumptions on these distributions. With a large sample-size, the minutiae of the two data generating processes are uncovered (e.g. slightly different calibration of the data collecting equipment, different numerical precision), and we ultimately reject the null hypothesis, even if the sources of variation across the two samples may be irrelevant for the analysis.

Similarly, we may be interested in *learning on distributions* [14, 23, 24], where the appropriate level of granularity in the data is distributional. For example, each label $y_i$ in supervised learning is associated to a whole bag of observations $B_i = \{X_{ij}\}_{j=1}^{N_i}$ – assumed to come from a probability distribution $P_i$, or we may be interested in clustering such bags of observations. Again, nonparametric distances used in such contexts to facilitate a learning algorithm on distributions, such as Maximum Mean Discrepancy (MMD) [5], can be sensitive to irrelevant sources of variation and may lead to suboptimal or even misleading results, in which case building predictors which are invariant to noise is of interest.

While it may be tempting to revert back to a parametric setup and work with simple, easy to interpret models, we argue that a different approach is possible: we stay within a nonparametric framework, exploit the irregular and complicated nature of real life distributions and *encode invariances* to sources

of variation assumed to be irrelevant. In this contribution, we focus on *invariances to symmetric additive noise* on each of the data generating distributions. Namely, assume that the $i$-th sample $\{X_{ij}\}_{j=1}^{N_i}$ we observe does not follow the distribution $P_i$ of interest but instead its convolution $P_i \star \mathcal{E}_i$ with some unknown noise distributions $\mathcal{E}_i$ assumed to be symmetric about 0 (we also require that it has a positive characteristic function). We would like to assess the differences between $P_i$ and $P_{i'}$ while allowing $\mathcal{E}_i$ and $\mathcal{E}_{i'}$ to differ in an arbitrary way. We investigate two approaches to this problem: (1) measuring the degree of asymmetry of the paired differences $\{X_{ij} - X_{i'j}\}$, and (2) comparing the *phase functions* of the corresponding samples. While the first approach is simpler and presents a sensible solution for the two-sample testing problem, we demonstrate that phase functions give a much better gauge on the *relative comparisons* between bags of observations, as required for learning on distributions.

The paper is outlined as follows. In section 2, we provide an overview of the background. In section 3, we provide details of the construction and implementation of phase features. In section 4, we discuss the approach based on asymmetry in paired differences for two sample testing with invariances. Section 5 provides experiments on synthetic and real data, before concluding in section 6.

## 2    Background and Setup

We will say that a random vector $E$ on $\mathbb{R}^d$ is a *symmetric positive definite (SPD) component* if its characteristic function is positive, i.e. $\varphi_E(\omega) = \mathbb{E}_{X \sim E}\left[\exp(i\omega^\top E)\right] > 0, \forall \omega \in \mathbb{R}^d$. This means that $E$ is (1) symmetric about zero, i.e. $E$ and $-E$ have the same distribution and (2) if it has a density, this density must be a positive definite function [20]. Note that many distributions used to model additive noise, including the spherical zero-mean Gaussian distribution, as well as multivariate Laplace, Cauchy or Student's $t$ (but not uniform), are all SPD components.

Following the terminology similar to that of [3], we will say that a random vector $X$ on $\mathbb{R}^d$ is *decomposable* if its characteristic function can be written as $\varphi_X = \varphi_{X_0}\varphi_E$, with $\varphi_E > 0$. Thus, if $X$ can be written in the form $X = X_0 + E$, where $X_0$ and $E$ are independent and $E$ is an SPD noise component, then $X$ is decomposable. We will say that $X$ is *indecomposable* if it is not decomposable. In this paper, we will assume that mostly the indecomposable components of distributions are of interest and will construct tools to directly measure differences between these indecomposable components, encoding invariance to other sources of variability. The class of Borel Probability measures on $\mathbb{R}^d$ will be denoted $\mathcal{M}_+^1(\mathbb{R}^d)$, while the class of indecomposable probability measures will be denoted by $\mathcal{I}(\mathbb{R}^d) \subseteq \mathcal{M}_+^1(\mathbb{R}^d)$.

### 2.1    Kernel Embeddings, Fourier Features and learning on distributions

For any positive definite function $k \colon \mathcal{X} \times \mathcal{X} \mapsto \mathbb{R}$, there exists a unique reproducing kernel Hilbert space (RKHS) $\mathcal{H}_k$ of real-valued functions on $\mathcal{X}$. Function $k(\cdot, x)$ is an element of $\mathcal{H}_k$ and represents evaluation at $x$, i.e. $\langle f, k(\cdot, x)\rangle_\mathcal{H} = f(x), \forall f \in \mathcal{H}_k, \forall x \in \mathcal{X}$. The kernel mean embedding (cf. [15] for a recent review) of a probability measure $P$ is defined by $\mu_P = \mathbb{E}_{X \sim P}[k(\cdot, X)] = \int_\mathcal{X} k(\cdot, x)dP(x)$. The Maximum Mean Discrepancy (MMD) between probability measures $P$ and $Q$ is then given by $\|\mu_P - \mu_Q\|_{\mathcal{H}_k}$. For shift-invariant kernels on $\mathbb{R}^d$, using Bochner's characterisation of positive definiteness [26, 6.2], the squared MMD can be written as a weighted $L_2$-distance between characteristic functions [22, Corollary 4]

$$\|\mu_P - \mu_Q\|_{\mathcal{H}_k}^2 = \int_{\mathbb{R}^d} |\varphi_P(\omega) - \varphi_Q(\omega)|^2 \, d\Lambda(\omega), \qquad (1)$$

where $\Lambda$ is the non-negative spectral measure (inverse Fourier transform) of kernel $k$ as a function of $x - y$, while $\varphi_P(\omega)$ and $\varphi_Q(\omega)$ are the characteristic functions of probability measures $P$ and $Q$.

Bochner's theorem is also used to construct random Fourier features (RFF) [19] for fast approximations to kernel methods in order to approximate a pre-specified shift-invariant kernel by a finite dimensional explicit feature map. If we can draw samples from its spectral measure $\Lambda$, we can

approximate $k$ by[1]

$$\hat{k}(x, y) = \frac{1}{m} \sum_{j=1}^{m} \left[ \cos(\omega_j^T x) \cos(\omega_j^T y) + \sin(\omega_j^T x) \sin(\omega_j^T y) \right] = \langle \phi(x), \phi(y) \rangle_{\mathbb{R}^{2m}}$$

where $\omega_1, \ldots, \omega_m \sim \Lambda$ and $\phi(x) := \sqrt{\frac{1}{m}} \left[ \cos\left(\omega_1^\top x\right), \sin\left(\omega_1^\top x\right) \ldots, \cos\left(\omega_m^\top x\right), \sin\left(\omega_m^\top x\right) \right]$. Thus, the explicit computation of the kernel matrix is not needed and the computational complexity is reduced. This also allows computation with the approximate, finite-dimensional embeddings $\tilde{\mu}_P = \Phi(P) = \mathbb{E}_{X \sim P} \phi(X) \in \mathbb{R}^{2m}$, which can be understood as the evaluations (real and complex part stacked together) of the characteristic function $\varphi_P$ at frequencies $\omega_1, \ldots, \omega_m$. We will refer to the approximate embeddings $\Phi(P)$ as Fourier features of distribution $P$.

Kernel embeddings can be used for supervised learning on distributions. Assume we have a training set $\{B_i, y_i\}_{i=1}^n$, where input $B_i = \{x_{ij}\}_{j=1}^{N_i}$ is a bag of samples taking values in $\mathcal{X}$, and $y_i$ is a response. Given a kernel $k \colon \mathcal{X} \times \mathcal{X} \to \mathbb{R}$, we first map each $B_i$ to the empirical embedding $\mu_{\hat{P}_i} = \frac{1}{N_i} \sum_{j=1}^{N_i} k(\cdot, x_{ij}) \in \mathcal{H}_k$ and then can apply any positive definite kernel on $\mathcal{H}_k$ as the kernel on bag inputs, e.g. linear kernel $\tilde{K}(B_i, B_i') = \langle \mu_{\hat{P}_i}, \mu_{\hat{P}_{i'}} \rangle_{\mathcal{H}_k}$, in order to perform classification [14] or regression [24]. Approximate kernel embeddings have also been applied in this context [23].

## 3 Phase Discrepancy and Phase Features

While MMD and kernel embeddings are related to characteristic functions, and indeed the same connection forms a basis for fast approximations to kernel methods using random Fourier features [19], the relevant notion in our context is the *phase function* of a probability measure, recently used for nonparametric deconvolution by [3]. In this section, we overview this formalism. Based on the empirical phase functions, we will then derive and investigate hypothesis testing and learning framework using *phase features of distributions*.

In nonparametric deconvolution [3], the goal is to estimate the density function $f_0$ of a univariate r.v. $X_0$, but in general we only have noisy data samples $X_1, \ldots, X_n \overset{iid}{\sim} X = X_0 + E$, where $E$ denotes an independent noise term. Even though the distribution of $E$ is unknown, making the assumption that $E$ is an SPD noise component, and that $X_0$ is indecomposable, i.e. $X_0$ itself does not contain any SPD noise components, [3] show that it is possible to obtain consistent estimates of $f_0$.

They distinguish between the symmetric noise and the underlying indecomposable component by matching phase functions, defined as

$$\rho_X(\omega) = \frac{\varphi_X(\omega)}{|\varphi_X(\omega)|}$$

where $\varphi_X(\omega)$ denotes the characteristic function of $X$. Observe that $|\rho_X(\omega)| = 1$, and thus we are effectively removing the amplitude information from the characteristic function. For a SPD noise component $E$, the phase function is $\rho_E(\omega) \equiv 1$. But then since $\varphi_X = \varphi_{X_0} \varphi_E$, we have that $\rho_{X_0} = \rho_X = \varphi_X / |\varphi_X|$, i.e. the phase function is invariant to additive SPD noise components. This motivates us to construct explicit feature maps of distributions with the same property and similarly to the motivation of [3], we argue that real-world distributions of interest often exhibit certain amount of irregularity and it is exactly this irregularity which is exploited in our methodology.

In analogy to the MMD, we first define the phase discrepancy (PhD) as a weighted $L_2$-distances between the phase functions:

$$\text{PhD}(X, Y) = \int_{\mathbb{R}^d} |\rho_X(\omega) - \rho_Y(\omega)|^2 \, d\Lambda(\omega) \tag{2}$$

for some non-negative measure $\Lambda$ (w.l.o.g. a probability measure). Now suppose we write $X = X_0 + U$, $Y = Y_0 + V$, where $U$ and $V$ are SPD noise components. This then implies $\rho_X = \rho_{X_0}$ and $\rho_Y = \rho_{Y_0}$ $\Lambda$-everywhere, so that $\text{PhD}(X, Y) = \text{PhD}(X_0, Y_0)$. It is clear then that the PhD is

not affected by additive SPD noise components, so it captures desired invariance. However, the PhD for $\Lambda$ supported everywhere is in fact not a proper metric on the indecomposable probability measures $\mathcal{I}(\mathbb{R}^d)$, as one can find indecomposable random variables $X$ and $Y$ s.t. $\rho_X = \rho_Y$ and thus $PhD(X, Y) = 0$. An example is given in Appendix A.

While such cases appear contrived, we hence restrict attention to a subset of indecomposable probability measures $\mathcal{P}(\mathbb{R}^d) \subset \mathcal{I}(\mathbb{R}^d)$, which are uniquely determined by phase functions, i.e. $\forall P, Q \in \mathcal{P}(\mathbb{R}^d) : \rho_P = \rho_Q \Rightarrow P = Q$.

We now have the two following propositions (proofs are given in Appendix B).

**Proposition 1.**

$$PhD(X, Y) = 2 - 2 \int \left( \frac{\mathbb{E}\xi_\omega(X)}{\|\mathbb{E}\xi_\omega(X)\|} \right)^\top \left( \frac{\mathbb{E}\xi_\omega(Y)}{\|\mathbb{E}\xi_\omega(Y)\|} \right) d\Lambda(\omega)$$

*where $\xi_\omega(x) = \left[ \cos\left(\omega^\top x\right), \sin\left(\omega^\top x\right) \right]^\top$ and $\| \cdot \|$ denotes the standard $L_2$ norm.*

**Proposition 2.**

$$K(P_X, P_Y) = \int \left( \frac{\mathbb{E}\xi_\omega(X)}{\|\mathbb{E}\xi_\omega(X)\|} \right)^\top \left( \frac{\mathbb{E}\xi_\omega(Y)}{\|\mathbb{E}\xi_\omega(Y)\|} \right) d\Lambda(\omega)$$

*is a positive definite kernel on probability measures.*

Now, we can construct an approximate explicit feature map for kernel $K$. Taking a sample $\{\omega_i\}_{i=1}^m \sim \Lambda$, we define $\Psi : P_X \mapsto \mathbb{R}^{2m}$ given by $\Psi(P_X) = \sqrt{\frac{1}{m}} \left[ \frac{\mathbb{E}\xi_{\omega_1}(X)}{\|\mathbb{E}\xi_{\omega_1}(X)\|}, \ldots, \frac{\mathbb{E}\xi_{\omega_m}(X)}{\|\mathbb{E}\xi_{\omega_m}(X)\|} \right]$. We will refer to $\Psi(\cdot)$ as the *phase features*. Note that these are very similar to Fourier features, but the $\cos, \sin$-pair corresponding to each frequency is normalised to have unit $L_2$ norm. In other words, $\Psi(\cdot)$ can be thought of as evaluations of the phase function at the selected frequencies. By construction, phase features are invariant to additive SPD noise components. For an empirical measure, we simply have the following:

$$\Psi(\hat{P}_X) = \sqrt{\frac{1}{m}} \left[ \frac{\hat{\mathbb{E}}\xi_{\omega_1}(X)}{\|\hat{\mathbb{E}}\xi_{\omega_1}(X)\|}, \ldots, \frac{\hat{\mathbb{E}}\xi_{\omega_m}(X)}{\|\hat{\mathbb{E}}\xi_{\omega_m}(X)\|} \right] \tag{3}$$

where we have replaced the expectations by their empirical estimates. Because $\left\|\Psi(\hat{P}_X)\right\| = 1$, we can construct

$$\widehat{PhD}(\hat{P}_X, \hat{P}_Y) = \left\| \Psi(\hat{P}_X) - \Psi(\hat{P}_Y) \right\|^2 = 2 - 2\Psi(\hat{P}_X)^\top \Psi(\hat{P}_Y), \tag{4}$$

which is a Monte Carlo estimator of $PhD(\hat{P}_X, \hat{P}_Y)$. In summary, $\Psi(\hat{P}) \in \mathbb{R}^{2m}$ is an explicit feature vector of the empirical distribution which encodes invariance to additive SPD noise components present in $P$ [2], as demonstrated in Figure F.1 in the Appendix. It can now be directly applied to (1) two-sample testing up to SPD components, where the distance between the phase features, i.e. an estimate (4) of the PhD, can be used as a test statistic, with details given in section 5.1 and (2) learning on distributions, where we use phase features as the explicit feature map for a bag of samples.

Although we have assumed an indecomposable underlying distribution so far, this assumption is not strict. For distribution regression, if the indecomposable assumption is invalid, given that the underlying distribution is irregular, it may still be useful to encode invariance as long as the benefit of removing the SPD components irrelevant for learning outweighs the signal in the SPD part of the distribution, i.e. there is a trade off between SPD noise and SPD signal. In practice, the phase features we propose can be used to encode such invariance where appropriate or in conjunction with other features which do not encode invariance.

In order to construct the approximate mean embeddings for learning, we first compute an explicit feature map by taking averages of the Fourier features, as given by $\Phi(\hat{P}_X) = \sqrt{\frac{1}{m}} \left[ \hat{\mathbb{E}}\xi_{\omega_1}(X), \ldots, \hat{\mathbb{E}}\xi_{\omega_m}(X) \right]$. For phase features, we need to compute an additional normalisation term over each frequency as in (3). To obtain the set of frequencies $\{w_i\}_{i=1}^m$, we can draw

samples from a probability measure $\Lambda$ corresponding to an inverse Fourier transform of a shift-invariant kernel, e.g. Gaussian Kernel. However, given a supervised signal, we can also optimise a set of frequencies $\{w_i\}_{i=1}^m$ that will give us a useful representation and good discriminative performance. In other words, we no longer focus on a specific shift-invariant kernel $k$, but are *learning discriminative Fourier/phase features*. To do this, we can construct a neural network (NN) with special activation functions, pooling layers as shown in Algorithm D.1 and Figure D.1 in the Appendix.

## 4   Asymmetry in Paired Differences

We now consider a separate approach to nonparametric two-sample test, where we wish to test the null hypothesis that $H_0 : P\overset{d}{=}Q$ vs. the general alternative, but we only have iid samples arising from $X \sim P \star \mathcal{E}_1$ and $Y \sim Q \star \mathcal{E}_2$. i.e.

$$X = X_0 + U \quad Y = Y_0 + V$$

where $X_0 \sim P$, $Y_0 \sim Q$ lie in the space of $\mathcal{P}(\mathbb{R}^d)$ of indecomposable distributions uniquely determined by phase functions and $U$ and $V$ are SPD noise components. With this setting (proof in Appendix B):

**Proposition 3.** *Under the null hypothesis $H_0$, $X - Y$ is SPD $\iff X_0\overset{d}{=}Y_0$.*

This motivates us to simply perform a two-sample test on $X - Y$ and $Y - X$ since its rejection would imply rejection of $X_0\overset{d}{=}Y_0$, as it tests for symmetry. However, note that this is a test for symmetry only and that for consistency against all alternatives, positivity of characteristic function would need to be checked separately. Now, given two i.i.d. samples $\{X_i\}_{i=1}^n$ and $\{Y_i\}_{i=1}^n$ with $n$ even, we split the two samples into two halves and compute $W_i = X_i - Y_i$ on one half and $Z_i = Y_i - X_i$ on the other half, and perform a nonparametric two sample test on $W$ and $Z$ (which are, by construction, independent of each other). The advantage of this regime is that we can use any two-sample test – in particular in this paper, we will focus on the *linear time* mean embedding (ME) test [7], which was found to have performance similar to or better than the original MMD two-sample test [5], and explicitly formulates a criterion which maximises the test power. We will refer to the resulting test on paired differences as the Symmetric Mean Embedding (SME).

Although we have assumed here that $X_0, Y_0$ lie in the space $\mathcal{P}(\mathbb{R}^d)$ of indecomposable distributions, in practice, the SME test would not reject if the underlying distributions of interest *differ only in the symmetric components* (or in the SPD components for the PhD test). We argue this to be unlikely due to real life distributions being complex in nature with interesting differences often having a degree of asymmetry. In practice, we recommend the use of the ME and SME or PhD test together to provide an exploratory tool to understand the underlying differences, as demonstrated in the Higgs Data experiment in section 5.1.

It is tempting to also consider learning on distributions with invariances using this formalism. However note that the MMD on paired differences is *not invariant to the additive SPD noise components* under the alternative, i.e. in general $\mathrm{MMD}(X - Y, Y - X) \neq \mathrm{MMD}(X_0 - Y_0, Y_0 - X_0)$. This means that the paired differences approach to learning is sensitive to the actual type and scale of the additive SPD noise components, hence not suitable for learning. The mathematical details and empirical experiments to show this are presented in Appendix C and F.1.

## 5   Experimental Results

### 5.1   Two-Sample Tests with Invariances

In this section, we demonstrate the performance of the SME test and the PhD test on both artificial and real-world data for testing the hypothesis $H_0 : X_0\overset{d}{=}Y_0$ based on samples $\{X_i\}_{i=1}^N$ from $X_0 + U$ and $\{Y_i\}_{i=1}^N$ from $Y_0 + V$, where $U$ and $V$ are arbitrary SPD noise components (we assume the same number of samples for simplicity). SME test follows the setup in [7] but applied to $\{X_i - Y_i\}_{i=1}^{N/2}$ and $\{Y_i - X_i\}_{i=N/2+1}^N$. For the PhD test, we use as the test statistic the estimate $\widehat{\mathrm{PhD}}(\hat{P}_X, \hat{P}_Y)$ of (2). It is unclear what the exact form of the null distribution is, so we use a permutation test, by recomputing this statistic on the samples which are first merged and then randomly split in the original proportions.

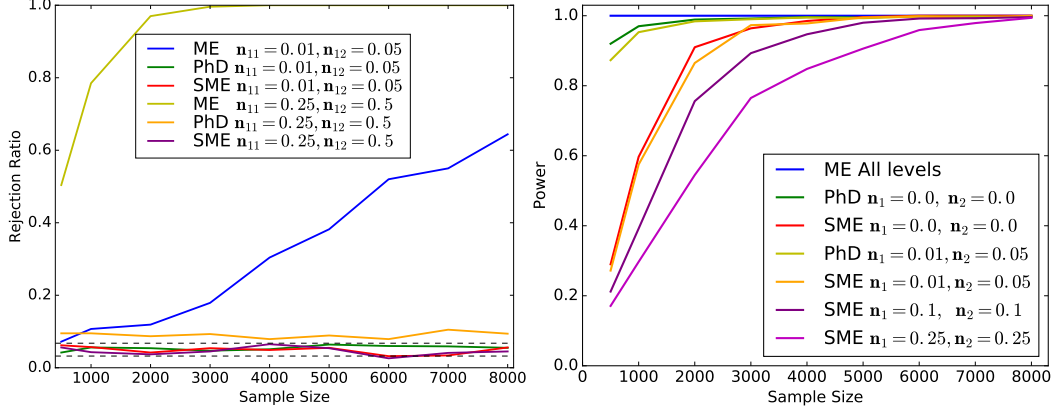

Figure 1: Type I error and Power under various additional symmetric noise in the synthetic $\chi^2$ dataset. Dashed line is the 99% Wald interval here. **Left:** Type I error, $n_{11}$ denotes the noise to signal ratio for the first set of samples and $n_{12}$ for the second set. **Right:** Power, $n_1$ denotes the noise to signal ratio for the $X$ set of samples and $n_2$ denotes the noise to signal ratio for the $Y$ set of samples.

While we are combining samples with different distributions, the permutation test is still justified since, under the null hypothesis $X_0 \overset{d}{=} Y_0$, the resulting characteristic function $\varphi_{null}$ of the mixture can be written as

$$\varphi_{null} = \frac{1}{2}\varphi_{X_0}\varphi_U + \frac{1}{2}\varphi_{X_0}\varphi_V = \varphi_{X_0}(\frac{1}{2}\varphi_U + \frac{1}{2}\varphi_V)$$

and since the mixture of the SPD noise terms is also SPD, we have that $\rho_{null} = \rho_{X_0} = \rho_{Y_0}$. For our experiments, we denote by $N$ the sample size, $d$ the dimension of the samples, and we take $\alpha = 0.05$ to be the significance level. In the SME test, we take the number of test locations $J$ to be 10, and use 20% of the samples to optimise the test locations. All experimental results are averaged over 1000 runs, where each run repeats the simulation or randomly samples without replacement from the dataset.

### 5.1.1   Synthetic example: Noisy $\chi^2$

We start by demonstrating our tests with invariances on a simulated dataset where $X_0$ and $Y_0$ are random vectors with $d = 5$, each dimension is the same in distribution and follows $\chi^2(4)/4$ and $\chi^2(8)/8$ respectively, i.e. chi-squared random variables, with different degrees of freedom, rescaled to have the same mean 1 (but have different variances, $1/2$ and $1/4$ respectively). An illustration of the true and empirical phase and characteristic function with noise for these two distributions can be found in Appendix F.2. We construct samples $\{X_{n_1,i}\}_{i=1}^N$ and $\{Y_{n_2,i}\}_{i=1}^N$ such that $X_{n_1} \sim X_0 + U$, where $U \sim \mathcal{N}(0, \sigma_1^2 I)$ and similarly $Y_{n_2} \sim Y_0 + V$, where $V \sim \mathcal{N}(0, \sigma_2^2 I)$, $n_i$ denotes the noise-to-signal ratio given by the ratio of variances in each dimension, i.e. $n_1 = 2\sigma_1^2$ and $n_2 = 4\sigma_2^2$.

We first verify that Type I error is indeed controlled at our design level of $\alpha = 0.05$ *up to various additive SPD noise components*. This is shown in Figure 1 (left), where $X_0 \overset{d}{=} Y_0$, both constructed using $\chi^2(4)/4$, with the noiseless case found in Figure F.6 in the Appendix. It is noted here that the ME test rejects the null hypothesis for even a small difference in noise levels, hence it is unable to let us *target the underlying distributions* we are concerned with. This is unlike the SME test which controls the Type I error even for large differences in noise levels. The PhD test, on the other hand, while correctly controlling Type I at small noise levels, was found to have inflated Type I error rates for large noise, with more results and explanation provided in Figure F.6 in the Appendix. Namely, the test relies on the invariance to SPD of the population expression of PhD, but the estimator of the null distribution of the corresponding test statistic will in general be affected by the differing noise levels.

Next, we investigate the power, shown in Figure 1 (right). For a fair comparison, we have included the PhD test power only for small noise levels, in which the Type I error is controlled at the design level. In these cases, the PhD test has better power than the SME test. This is not surprising, as for the SME we have to halve the sample size in order to construct a valid test. However, recall that the PhD test has an inflated Type I error for large noises, which means that its results should be considered with caution in practice. ME test rejects at all levels at all sample sizes as it picks up all possible

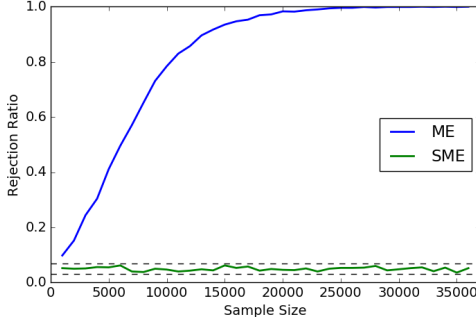

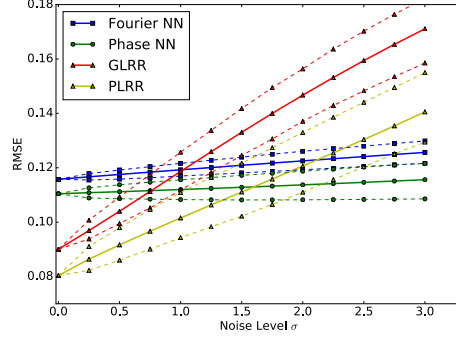

Figure 2: Rejection ratio vs. sample size for extremely low level features for Higgs dataset. Dashed line is the 99% Wald interval for 1000 repetitions for $\alpha = 0.05$. Note PhD is not used here, due to its expensive computational cost.

Figure 3: RMSE on the Aerosol test set, corrupted by various levels of noise averaged over 100 runs, with the $5^{th}$ and the $95^{th}$ percentile. The noiseless case is shown with one run. RMSE from mean is $0.206$.

differences. SME and PhD are by construction more conservative tests whose rejection provides a much stronger statement: two samples differ even when *all arbitrary additive SPD components* have been stripped off.

### 5.1.2 Higgs Dataset

The UCI Higgs dataset [1, 11] is a dataset with 11 million observations, where the problem is to distinguish between the signal process where Higgs bosons are found, versus the background process that do not produce Higgs bosons. In particular, we will consider a two-sample test with the ME and SME test on the high level features derived by physicists, as well as a two-sample test on four extremely low level features (azimuthal angular momentum $\phi$ measured by four particle jets in the detector). The high level features here (in $\mathbb{R}^7$) have been shown to have good discriminative properties in [1]. Thus, we expect them to have different distributions across two processes. Denoting by $X$ the high level features of the process without Higgs Boson, and $Y$ as the corresponding distribution for the processes where Higgs bosons are produced, we test the null hypothesis that the indecomposable parts of $X$ and $Y$ agree. The results can be found in Table F.1 in the Appendix, which shows that the high level features differ even up to additive SPD components, with a high power for the SME and ME test even at small sample sizes (rejection rate of $0.94$ at $N = 500$). Now we perform the same experiment, but with the low level features $\in \mathbb{R}^4$, commented in [1] to carry very little discriminating information, using the setup from [2].

The results for the ME and SME test can be found in Figure 2. Here we observe that while ME test clearly rejects and finds the difference between the two distributions, there is no evidence that the indecomposable parts of the joint distributions of the angular momentum actually differ. In fact, the test rejection rate remains around the chosen design level of $\alpha = 0.05$ for all sample sizes. This highlights the significance in using the SME test, suggesting that the nature of the difference between the two processes can potentially be explained by some additive symmetric noise components which may be irrelevant for discrimination, providing an insight into the dataset. Furthermore, this also highlights the argument that given two samples from complex data collection and generation processes, a nonparametric two sample test like ME will likely reject given sufficient sample sizes, even if the discovered difference may not be of interest. With the SME test however, we can ask a much more subtle question about the differences between the assumed true underlying processes. Figures showing that the Type I error is controlled at the design level of $\alpha = 0.05$ for both low and high level features can be found in Figure F.7 in the Appendix.

## 5.2 Learning with Phase Features

### 5.2.1 Aerosol Dataset

To demonstrate the phase features invariance to SPD noise component, we use the Aerosol MISR1 dataset also studied by [24] and [25] and consider a situation with *covariate shift* [18] on distribution inputs: the testing data is impaired by additive SPD components different to that in the training data.

Table 1: Mean Square Error (MSE) on dark matter dataset for $500$ runs with $5^{th}$ and $95^{th}$ percentile.

| Algorithm | MSE |
|---|---|
| Mean | 0.16 |
| PLRR | **0.021** $(0.018, 0.024)$ |
| GLRR | 0.033 $(0.030, 0.037)$ |
| LGRR | 0.032 $(0.028, 0.036)$ |
| PGRR | 0.021 $(0.017, 0.024)$ |
| GGRR | **0.018** $(0.015, 0.019)$ |

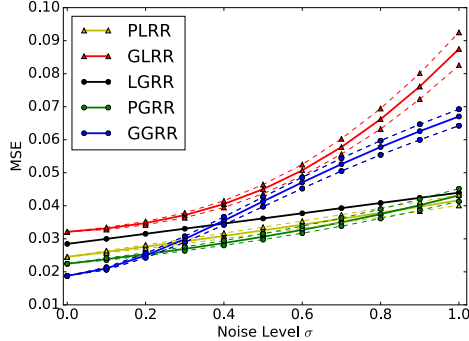

Figure 4: MSE with various levels of noise added on test set, with $5^{th}$ and $95^{th}$ percentile.

Here, we have an aerosol optical depth (AOD) multi-instance learning problem with 800 bags, where each bag contains 100 randomly selected multispectral (potentially cloudy) pixels within 20km radius around an AOD sensor. The label $y_i$ for each bag is given by the AOD sensor measurements and each sample $x_i$ is 16-dimensional. This can be understood as a distribution regression problem where each bag is treated as a set of samples from some distribution.

We use 640 bags for training and 160 bags for testing. Here in the bags for testing *only*, we add varying levels of Gaussian noise $\epsilon \sim \mathcal{N}(0, Z)$ to each bag, where $Z$ is a diagonal matrix with diagonal components $z_i \sim U[0, \sigma v_i]$ with $v_i$ being the empirical variance in dimension $i$ across all samples, accounting for different scales across dimensions. For comparisons, we consider linear ridge regression on embeddings with respect to a Gaussian kernel, approximated with RFF (GLRR) as described in section 2.1 (i.e. a linear kernel is applied on approximate embeddings), linear ridge regression on phase features (PLRR) (i.e. normalisation step is applied to obtain (3)), and also the phase and Fourier neural networks (NN), described in Appendix D, tuning all hyperparameters with 3-fold cross validation. With the same model, we now measure Root Mean Square Error (RMSE) 100 times with various noise-corrupted test sets and results are shown in figure 3. It is also noted that a second level non-linear kernel $\tilde{K}$ does not improve performance significantly on this problem [24].

We see that GLRR and PLRR are competitive (see Appendix Table F.2) in the noiseless case, and these clearly outperform both the Fourier NN and Phase NN (likely due to the small size of the dataset). For increasing noise, the performance of GLRR degrades significantly, and while the performance of PLRR degrades also, the model is much more robust under additional SPD noise. In comparison, the Phase NN implementation is almost insensitive to covariate shift in the test sets, unlike the performance of PLRR, highlighting the importance of learning discriminative frequencies $w$ in a very low signal-to-noise setting.

It is noted that the Fourier NN performs similarly to that of the Phase NN on this example. Interestingly, discriminative frequencies learnt on the training data correspond to Fourier features that are nearly normalised (i.e. they are close to unit norm - see Figure F.8 in the Appendix). This means that the Fourier NN has *learned to be approximately invariant* based on training data, indicating that the original Aerosol data potentially has irrelevant SPD noise components. This is reinforced by the nature of the dataset (each bag contains 100 randomly selected potentially cloudy pixels, known to be noisy [25]) and no loss of performance from going from GLRR to PLRR. The results highlights that phase features are stable under additive SPD noise.

### 5.2.2 Dark Matter Dataset

We now study the use of phase features on the dark matter dataset, composing of a catalog of galaxy clusters. In this setting, we would like to predict the total mass of galaxy clusters, using the dispersion of velocities in the direction along our line of sight. In particular, we will use the 'ML1' dataset, as obtained from the authors of [16, 17], who constructed a catalog of massive halos from the MultiDark `mdpl` simulation [9]. The dataset contains 5028 bags, with each sample consisting of its sub-object velocity and its mass label in $\mathbb{R}$. By viewing each galaxy cluster at multiple lines of sights, we obtain 15 000 bags, using the same experimental setup as in [10]. For experiments, we use approximately 9000 bags for training, and 3000 bags each for validation and testing, keeping those of multiple lines of sight in the same set. As before, we use GLRR and PLRR and we also include

in comparisons methods with a second level Gaussian kernel (with RFF) applied to phase features (PGRR) and to approximate embeddings (GGRR). For a baseline, we also include a first level linear kernel (equivalent to representing each bag with its mean), before applying a second level gaussian kernel (LGRR). We use the same set of randomly sampled frequencies across the methods, tuning for the scale of the frequencies and for regularisation parameters.

Table 1 shows the results of the methods across 10 different data splits, with 50 sets of randomised frequencies for each data split. We see that PLRR is significantly better than GLRR. This suggests that under this model structure, by removing SPD components from each bag, we can target the underlying signal and obtain superior performance, highlighting the applicability of phase features. Considering a second level gaussian kernel, we see that the GGRR has a slight advantage over PGRR, with PGRR performing similar to PLRR. This suggests that the SPD components of the distribution of sub-object velocity may be useful for predicting the mass of a galaxy cluster if an additional nonlinearity is applied to embeddings – whereas the benefits of removing them outweigh the signal present in them without this additional nonlinearity. To show that indeed the phase features are robust to SPD components, we perform the same covariate shift experiment as in the aerosol dataset, with results given in Figure 4. Note that LGRR is robust to noise, as each bag is represented by its mean.

## 6    Conclusion

No dataset is immune from measurement noise and often this noise differs across different data generation and collection processes. When measuring distances between distributions, can we disentangle the differences in noise from the differences in the signal? We considered two different ways to encode invariances to additive symmetric noise in those distances, each with different strengths: a nonparametric measure of asymmetry in paired sample differences and a weighted distance between the empirical phase functions. The former was used to construct a hypothesis test on whether the difference between the two generating processes can be explained away by the difference in postulated noise, whereas the latter allowed us to introduce a flexible framework for invariant feature construction and learning algorithms on distribution inputs which are robust to measurement noise and target underlying signal distributions.

## Acknowledgements

We thank Dougal Sutherland for suggesting the use of of the dark matter dataset, Michelle Ntampaka for providing the catalog, as well as Ricardo Silva, Hyunjik Kim and Kaspar Martens for useful discussions. This work was supported by the EPSRC and MRC through the OxWaSP CDT programme (EP/L016710/1). C.Y. and H.C.L.L. also acknowledge the support of the MRC Grant No. MR/L001411/1.

The CosmoSim database used in this paper is a service by the Leibniz-Institute for Astro-physics Potsdam (AIP). The MultiDark database was developed in cooperation with the Spanish MultiDark Consolider Project CSD2009-00064. The authors gratefully acknowledge the Gauss Centre for Supercomputing e.V. (www.gauss-centre.eu) and the Partnership for Advanced Supercomputing in Europe (PRACE, www.prace-ri.eu) for funding the MultiDark simulation project by providing computing time on the GCS Supercomputer SuperMUC at Leibniz Supercomputing Centre (LRZ, www.lrz.de).

## Footnotes

[1] a *complex feature map* $\phi(x) = \sqrt{\frac{1}{m}} \left[ \exp\left(i \omega_1^\top x\right), \ldots, \exp\left(i \omega_m^\top x\right) \right]$ can also be used, but we follow the convention of real-valued Fourier features, since kernels of interest are typically real-valued.

[2]Note that, unlike the population expression $\Psi(P)$, the empirical estimator $\Psi(\hat{P})$ will in general have a distribution affected by the noise components and is thus only approximately invariant, but we observe that it captures invariance very well as long as the signal-to-noise regime remains relatively high (Section 5.1).

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
