[Supplementary Material · nips_camera_ready_appendix.pdf]

# A Different Indecomposable Distributions Can Coincide in Phase

Let $X$ and $Y$ be (univariate) random variables with densities

$$f_X(x) = \frac{1}{\sqrt{2\pi}} x^2 \exp(-x^2/2), \quad f_Y(x) = \frac{1}{2}|x| \exp(-|x|).$$

Then it can be directly checked that their characteristic functions are given by

$$\varphi_X(\omega) = (1 - \omega^2) \exp(-\omega^2/2), \quad \varphi_Y(\omega) = \frac{1 - \omega^2}{(1 + \omega^2)^2}.$$

Thus, the phase functions coincide and are equal to

$$\rho_X(\omega) = \rho_Y(\omega) = \begin{cases} +1, |\omega| < 1, \\ -1, |\omega| > 1, \\ \text{undefined}, \omega \in \{-1, 1\}. \end{cases}$$

However, it is can also checked that even though they are symmetric, $X$ and $Y$ are indecomposable, cf. e.g. [12], which use a related but distinct notion of indecomposability of random variables. The plots of the densities and characteristic functions of $X$ and $Y$ are given in Fig. A.1.

Figure A.1: Example of two indecomposable distributions which have the same phase function. **Left**: densities. **Right**: charactersitic functions.

# B Phase Discrepancy and Asymmetry in Paired Differences Proofs

In this section, we will provide further details of the definitions, calculations and proofs in section 3 and 4. Phase discrepancy is defined as the weighted $L_2$-distances between the phase functions, i.e.

$$\mathrm{PhD}(X, Y) = \int |\rho_X(\omega) - \rho_Y(\omega)|^2 \, d\Lambda(\omega),$$

for some positive measure $\Lambda$ (w.l.o.g. a probability measure). Phase discrepancy measures how much $X$ and $Y$ differ up to an independent SPD noise component. We note that while the form of the PhD is motivated by that of MMD (weighted $L_2$-distances between the characteristic functions), relating it to the properties of the corresponding kernel and its RKHS is not straightforward. For example, constructing a PhD interpretation as a supremum over the RKHS unit ball (which is often how MMD is introduced) is immediate only for the case where indecomposable parts are point masses. Namely, if $X = x_0 + U$ and $Y = y_0 + V$, i.e. indecomposable parts are almost surely constant vectors $x_0$ and $y_0$, then

$$PhD(X, Y) = \|k(\cdot, x_0) - k(\cdot, y_0)\|_{\mathcal{H}_k}^2 = \sup_{\|f\|_{\mathcal{H}_k} \leq 1} |f(x_0) - f(y_0)|^2,$$

by virtue of $\rho_X(\omega) = e^{i\omega^\top x_0} = \varphi_{x_0}(\omega)$. In other cases, while it is clear that the spectral properties of the kernel still regulate the amount of frequency content that is used, one obtains the RKHS distance between the kernel convolutions of the inverse Fourier transforms of the phase functions so the interpretation is less clear.

Below, we provide the proofs of the propositions from the main text.

**Proposition 4.**

$$PhD(X,Y) = 2 - 2 \int \frac{\mathbb{E}\cos\left(\omega^\top (X-Y)\right)}{\sqrt{\mathbb{E}\cos\left(\omega^\top (X-X')\right)\mathbb{E}\cos\left(\omega^\top (Y-Y')\right)}} d\Lambda(\omega).$$

*Proof.*

$$
\begin{aligned}
\mathrm{PhD}(X,Y) &= \int |\rho_X(\omega) - \rho_Y(\omega)|^2 \, d\Lambda(\omega) \\
&= \int |\rho_X(\omega)|^2 \, d\Lambda(\omega) + \int |\rho_Y(\omega)|^2 \, d\Lambda(\omega) - \int (\rho_X\overline{\rho_Y} + \overline{\rho_X}\rho_Y) \, d\Lambda \\
&= 2 - \int \frac{\varphi_X\overline{\varphi_Y} + \overline{\varphi_X}\varphi_Y}{|\varphi_X|\,|\varphi_Y|} d\Lambda \\
&= 2 - 2 \int \frac{\varphi_Z}{\sqrt{\varphi_{X-X'}\varphi_{Y-Y'}}} d\Lambda,
\end{aligned}
$$

where $X$ and $X'$ are iid, $Y$ and $Y'$ are iid and $Z$ is an equal mixture of $X-Y$ and $Y-X$. Indeed,

$$\varphi_X\overline{\varphi_Y} + \overline{\varphi_X}\varphi_Y = \varphi_{X-Y} + \varphi_{Y-X} = 2\varphi_Z,$$

and

$$\varphi_{X-X'} = \varphi_X\overline{\varphi_X} = |\varphi_X|^2.$$

Note that $X - X', Y - Y'$ and $Z$ are all symmetric. Thus,

$$
\begin{aligned}
\varphi_Z(\omega) &= \mathbb{E}\left[\cos\left(\omega^\top Z\right)\right] = \frac{1}{2}\mathbb{E}\left[\cos\left(\omega^\top (X-Y)\right)\right] + \frac{1}{2}\mathbb{E}\left[\cos\left(\omega^\top (Y-X)\right)\right] \\
&= \mathbb{E}\left[\cos\left(\omega^\top (X-Y)\right)\right].
\end{aligned}
$$

Substituting provides us the result. $\qquad\square$

**Proposition 5.** $K_\omega\left(\mathsf{P}_X, \mathsf{P}_Y\right) = \left(\frac{\mathbb{E}\xi_\omega(X)}{\|\mathbb{E}\xi_\omega(X)\|}\right)^\top \left(\frac{\mathbb{E}\xi_\omega(Y)}{\|\mathbb{E}\xi_\omega(Y)\|}\right)$ *is a positive definite kernel on probability measures* $\forall \omega$*, where here* $\xi_\omega(x) = \left[\cos\left(\omega^\top x\right), \sin\left(\omega^\top x\right)\right]$*, and so is* $K\left(\mathsf{P}_X, \mathsf{P}_Y\right) = \int K_\omega\left(\mathsf{P}_X, \mathsf{P}_Y\right) d\Lambda(\omega)$ *for any positive measure* $\Lambda$.

*Proof.* Define a feature map $\xi_\omega : \mathcal{X} \to \mathbb{R}^2$ with $\xi_\omega(x) = \left[\cos\left(\omega^\top x\right), \sin\left(\omega^\top x\right)\right]$, which induces a kernel on $\mathcal{X}$ given by $k_\omega(x,y) = \cos\left(\omega^\top (x-y)\right)$. Then $\kappa_\omega\left(\mathsf{P}_X, \mathsf{P}_Y\right) = \mathbb{E}\cos\left(\omega^\top (X-Y)\right) = \mathbb{E}k_\omega(X,Y) = (\mathbb{E}\xi_\omega(X))^\top \mathbb{E}\xi_\omega(Y)$ is a valid kernel on probability measures and so is the normalised kernel

$$K_\omega\left(\mathsf{P}_X, \mathsf{P}_Y\right) = \frac{\kappa_\omega\left(\mathsf{P}_X, \mathsf{P}_Y\right)}{\sqrt{\kappa_\omega\left(\mathsf{P}_X, \mathsf{P}_X\right)\kappa_\omega\left(\mathsf{P}_Y, \mathsf{P}_Y\right)}} = \left(\frac{\mathbb{E}\xi_\omega(X)}{\|\mathbb{E}\xi_\omega(X)\|}\right)^\top \left(\frac{\mathbb{E}\xi_\omega(Y)}{\|\mathbb{E}\xi_\omega(Y)\|}\right),$$

where we used that $\mathbb{E}\cos\left(\omega^\top (X-X')\right) = (\mathbb{E}\xi_\omega(X))^\top \mathbb{E}\xi_\omega(X') = \|\mathbb{E}\xi_\omega(X)\|^2$. For the last claim, simply note that integrating through the positive measure preserves positive semidefinitess, i.e. $\sum \alpha_i\alpha_j K(\mathsf{P}_i, \mathsf{P}_j) = \int \left(\sum \alpha_i\alpha_j K_\omega(\mathsf{P}_i, \mathsf{P}_j)\right) d\Lambda(\omega) \geq 0$. $\qquad\square$

As a direct corollary,

**Proposition 6.** $PhD(X,Y) = 2 - 2K\left(\mathsf{P}_X, \mathsf{P}_Y\right) = 2\int \left(1 - \left(\frac{\mathbb{E}\xi_\omega(X)}{\|\mathbb{E}\xi_\omega(X)\|}\right)^\top \left(\frac{\mathbb{E}\xi_\omega(Y)}{\|\mathbb{E}\xi_\omega(Y)\|}\right)\right) d\Lambda(\omega).$

**Proposition 7.** *Under the null hypothesis,* $X - Y$ *is SPD* $\iff X_0 \overset{d}{=} Y_0$.

*Proof.* Under $H_0$, since $X_0$ has the same distribution as $Y_0$, then so do $X - Y = X_0 - Y_0 + U - V$ and $Y - X = Y_0 - X_0 + V - U$ as $U - V$ is symmetric. Moreover, $\varphi_{X-Y} = |\varphi_{X_0}|^2\varphi_U\varphi_V > 0$, so $X - Y$ is SPD. Conversely, if we assume that $X - Y$ is SPD, i.e. $\varphi_X\overline{\varphi_Y} > 0$, then $\rho_{X_0}\overline{\rho_{Y_0}} > 0$. Since $|\rho_{X_0}| = |\rho_{Y_0}| = 1$, this implies that $\rho_{X_0} = \rho_{Y_0}$, and hence $X_0 \overset{d}{=} Y_0$, since we assumed that $X_0$ and $Y_0$ belong to $\mathcal{P}(\mathbb{R}^d)$. Hence, we have that $X - Y$ is SPD $\iff X_0 \overset{d}{=} Y_0$. $\qquad\square$

## C   Paired Differences

Another way to measure asymmetry of the difference between random vectors $X$ and $Y$ is to use $\mathrm{MMD}(X - Y, Y - X)$ instead of $\mathrm{PhD}(X, Y)$. However, this quantity is not invariant, i.e., $\mathrm{MMD}(X - Y, Y - X) \neq \mathrm{MMD}(X_0 - Y_0, Y_0 - X_0)$, and in fact the values will heavily depend on the distributions of $U$ and $V$. We note that

$$\varphi_{X-Y}(\omega) - \varphi_{Y-X}(\omega) \;\; = \;\; 2i\mathbb{E}\sin\left(\omega^\top (X - Y)\right),$$

so that we are effectively measuring the size of the imaginary part of the characteristic function of $X - Y$ (which should not be there if it is symmetric). There are several different ways in which we can write this quantity:

$$
\begin{aligned}
\mathrm{MMD}(X - Y, Y - X) \;\; &= \;\; \left\| \mathbb{E}k(\cdot, X - Y) - \mathbb{E}k(\cdot, Y - X) \right\|_{\mathcal{H}_k}^2 \\
&= \;\; \int \left| \varphi_X \overline{\varphi_Y} - \overline{\varphi_X}\varphi_Y \right|^2 d\Lambda \\
&= \;\; 4\int \left[ \mathbb{E}\sin\left(\omega^\top (X - Y)\right) \right]^2 d\Lambda(\omega) \\
&= \;\; \int |\varphi_X|^2 |\varphi_Y|^2 \left( 2 - \frac{\varphi_X \overline{\varphi_Y}}{\overline{\varphi_X}\varphi_Y} - \frac{\overline{\varphi_X}\varphi_Y}{\varphi_X \overline{\varphi_Y}} \right) d\Lambda.
\end{aligned}
$$

The last expression indicates that this quantity is affected by the amplitude of the individual characteristic functions, experimental details to show this empirically can be found in F.1. Moreover, the quantity does not appear to lend itself to the *feature on distributions* formalism, i.e. we were unable to derive some Hilbert space features $\Upsilon(\mathsf{P}) \in \mathcal{H}$ such that $\mathrm{MMD}(X - Y, Y - X) = \|\Upsilon(\mathsf{P}_X) - \Upsilon(\mathsf{P}_Y)\|_{\mathcal{H}}^2$, and it is thus unclear whether this approach can be used to define a valid kernel on distributions.

## D   Learning Discriminative Features

---
**Algorithm D.1** Phase/Fourier Neural Network

---
**Input:** Batch of bag of samples $X \in \mathbb{R}^{b \times N \times p}$, where $b$ is the batch size, $N$ is the bag size and $p$ is the dimension
**Output:** Classification or Regression Output
**1.** Compute $f(X) = XW$ where $W \in \mathbb{R}^{p \times m}$
**2.** Apply a $\sin$ and $\cos$ activation function

$$l_1(X) = [\sin(f(X)) \cos(f(X))]$$

**3.** Apply mean pooling operation over $N$, effectively computing $\hat{\mathbb{E}}\xi_{\omega_i}(X)$ for each $\omega_i \in \mathbb{R}^p$

$$l_2(X) = \left[ \hat{\mathbb{E}}\xi_{\omega_1}(X), \ldots, \hat{\mathbb{E}}\xi_{\omega_m}(X) \right] \in \mathbb{R}^{2m}$$

**4.** For Phase Neural Network, compute $\left\| \hat{\mathbb{E}}\xi_{\omega_1}(X) \right\|$ for each frequency and normalise to obtain:

$$l_3(X) = \left[ \frac{\hat{\mathbb{E}}\xi_{\omega_1}(X)}{\|\hat{\mathbb{E}}\xi_{\omega_1}(X)\|}, \ldots, \frac{\hat{\mathbb{E}}\xi_{\omega_m}(X)}{\|\hat{\mathbb{E}}\xi_{\omega_m}(X)\|} \right]$$

**5.** Batch Normalisation Layer
**6.** Output layer

---

Figure D.1: Main structure of the phase neural network

Algorithm D.1 shows the phase Neural Network (phase NN) and the Fourier Neural Network (Fourier NN), where the latter can be obtained by simply removing step 4 in the algorithm. Although the batch normalisation is not required, it is highly recommended for faster training of the network [6], due to the normalisation for the phase neural network in step 5 of the algorithm. Because of the neural network structure, we can take advantage of the rich literature, as well as alter the network in order to target a variety of different problems. For example, setting now the loss function as the squared loss, cross entropy or pinball loss, we can solve tasks in regression, classification or quantile regression on distributional inputs with discriminative frequencies. The Fourier neural network can also be extended to inputs in $\mathbb{R}^p$ for normal regression and classification problems by removing the mean pooling operation in step 3 of the algorithm.

## E   Distribution Regression with Invariance for ABC

---
**Algorithm E.1** Phase Regression, Fourier Regression
---

**Input:** prior $\pi$ for $\theta$, data-generating process $P$, phase or Fourier features
**Output:** Phase or Fourier Regression Neural Network
**for** $i = 1, \ldots, n$ **do**
    Sample $\theta_i \sim \pi$
    Sample dataset $B_i = \{x_{ij}\}_{j=1}^N$ from $P(\cdot|\theta_i)$
**end for**
Train Phase or Fourier neural network with $\{B_i, y_i\}_{i=1}^n$

---

---
**Algorithm E.2** Phase-ABC or Fourier-ABC
---

**Input:** prior $\pi$ for $\theta$, data-generating process $P$, observed data $B^* = \{x_j^*\}_{j=1}^{N^*}, \epsilon$, number of particles $K$
**Output:** Weighted Posterior sample $\sum_k w_k \delta_{\theta_k}$
**1.** Perform Phase or Fourier Regression, obtain $m(\cdot)$
**2.** ABC
**for** $k = 1, \ldots, K$ **do**
    Sample $\theta_k \sim \pi$
    Sample dataset $B_k = \{x_{kj}\}_j$ from $P(\cdot|\theta_k)$
    Compute $\widetilde{w}_k = \exp\left(-\dfrac{||m(B_k) - m(B^*)||_2^2}{\epsilon}\right)$
**end for**
$w_k = \widetilde{w}_k / \sum_k \widetilde{w}_k$

---

We have designed an explicit feature map for a bag of samples that can be used for any distribution regression problem. We now present its potential application to Approximate Bayesian Computation (ABC). Motivated by the approach of [4] and [13], we propose to use the phase features to construct an optimal summary statistic (under some loss function) for ABC. ABC is a Bayesian framework that allows us to approximate the posterior distribution of some parameter $\theta$ by approximating the likelihood function through simulations. To capture this approximation of the likelihood function, simulated datasets from the model are compared with the observed data using some lower dimensional summary statistics. If the summary statistic is sufficient, then there is no loss of information when projecting the data onto lower dimensional space. In practice however, sufficient statistics are not available for complex models of interest and instead using the strategy of [4], one can construct summary statistics that provide inference of $\theta$ which is optimal with respect to a given loss function.

In particular, we will focus on the squared loss function as given by $L(\theta, \theta') = (\theta - \theta')^2$. [4] showed that under this loss, the posterior mean of the $\theta$ given observations $\mathbf{X}$ is in fact the optimal summary statistic of $\mathbf{X}$ for the ABC procedure. However, since this quantity can not be analytically computed, one approach is to estimate it by fitting a regression model from simulated data, some examples of this include the semi-automatic ABC [4] and DR-ABC [13]. Here we focus

on ideas from DR-ABC, which uses a kernel distribution regression approach, treating each simulated dataset (given $\theta$ simulated from the prior) as a bag of samples and taking its label to be $\theta$. After training the regression model, it proceeds to using it as a summary statistic as given in algorithm E.2. The DR-ABC paper further proposed the conditional DR-ABC (CDR-ABC), which makes the assumption that only certain aspects of the data have an influence on $\theta$. By conditioning on such nuisance variables and then using conditional distribution regression (by embedding conditional distributions [21]), it can better account for the functional relationship inside the model. However, one problem with this approach is that the nuisance variables have to be observed directly, even for the true dataset, which may often not be the case. For example, consider the hierarchical model we used to illustrate the utility of phase features for regression below.

$$
\theta \sim \Gamma(\alpha, \beta), \quad Z \quad \sim \quad U[0, \sigma], \quad \epsilon \sim \mathcal{N}(0, Z),
$$
$$
X \quad \sim \quad \frac{\Gamma(\theta/2, 1/2)}{2\theta} + \epsilon, \tag{5}
$$

for some fixed values of $\alpha, \beta$ and $\sigma$. Here $\theta$ is the parameter we are interested in, $\epsilon$ is a latent noise variable (unobserved) and $X$ is the observation. Since neither $\epsilon$ nor $Z$ are observed on the true dataset, we can only use DR-ABC, not CDR-ABC. But DR-ABC then does not take into account the model structure which tells us that $\epsilon$ is irrelevant for inferring $\theta$, and it is thus likely to give poor performance for large values of $\sigma$. Hence, we propose to use phase features inside such regression model, which will be invariant to the noise variable $\epsilon$ which is an SPD component in observations. By using phase features for distribution regression, we should be able to better capture the functional relationship between $\theta$ and its corresponding dataset, a bag from $X|\theta$ and hence build better summary statistics for ABC. In some sense, this approach can be thought of as implicitly conditioning out the latent nuisance variable $\epsilon$, similarly as CDR-ABC does when it is observed. Furthermore, although we have chosen this example as an illustration, the phase features could be applied to many complex models with nuisance latent variables, even when we cannot write their contribution explicitly as here. The algorithms E.1 and E.2 shows the approach as in DR-ABC, but now replaced by our phase or Fourier regression approaches to compute summary statistics, and we denote these as Phase-ABC and Fourier-ABC.

## F  Additional Results

### F.1  Asymmetry in Paired Differences Experiment

Figure F.1: Histograms on various estimates for all pairs of bags with varying additive noise, red line denotes the noiseless case. **Top:** Estimated MMD on paired differences for all pair of bags, the red line given by the mean of the estimated MMD on paired differences for bags without noise. **Middle:** the squared distance between Fourier features (an estimate of MMD). **Bottom:** the squared distance between phase features (an estimate of PhD).

While it performed well when testing the null hypothesis, the MMD on paired differences is not invariant to the additive SPD noise components under the alternative hypothesis. Using the synthetic experimental setup as before, we simulate 100 noiseless bags from the two scaled $\chi^2$-distributions $X_0 \sim \chi^2(4)/4$ and $Y_0 \sim \chi^2(8)/8$, where each bag contains 1000 samples. We add varying levels of Gaussian noise to each bag, i.e. the bags are of the form $X_i = X_0 + \mathcal{N}(0, Z_i)$ and $Y_i = Y_0 + \mathcal{N}(0, W_i)$, where $Z_i, W_i \sim U[0, 0.1]$. We compute the estimate of the MMD on paired differences, the squared distance between Fourier features (an estimate of MMD) and the squared distance between phase features (an estimate of PhD) for all pairs of bags. In all computations, we used the same set of frequencies $\{w_i\}_{i=1}^{100}$ (sampled from a Gaussian distribution). We do the same for the noiseless samples (or use analytic expressions where available). The results are shown in figure F.1. We see that the MMD on paired differences is not invariant to SPD noise components (clearly, the noiseless case indicated by the red line has a much higher level of asymmetry than the noisy case where due to the presence of high levels of symmetric noise, differences often do appear symmetric). This is unlike the phase features, which maintain some level of invariance, the estimates stay away from 0 – preserving the signal about the difference of indecomposable $\chi^2$ components – and the mode is nearer the true value, even though there is clearly some variance, however this is expected as its PhD population expression is invariant, but not its estimator, furthermore the frequencies are sampled (with the median heuristic bandwidth) and not learnt. This suggests that phase features are more suitable for invariant learning on distributions than MMD on paired differences. The Fourier features are also given for comparison, but these are not expected to be invariant, as shown.

## F.2 Characteristic and Phase Function Plots

Figure F.2: The black line here correspond to the real and imaginary part of the true characteristic function of the $\chi^2(4)/4$ and $\chi^2(8)/8$ distribution, denoted $X, Y$ on the top and bottom graphs respectively. The red points denote the empirical characteristic function constructed with 750 frequencies sampled from a Gaussian kernel with $\sigma = 2$ using a bag size of 1000 observations, with some additional Gaussian noise.

Figure F.3: The black line here correspond to the real and imaginary part of the true phase function of the $\chi^2(4)/4$ and $\chi^2(8)/8$ distribution, denoted $X, Y$ on the top and bottom graphs respectively. The red points denote the empirical phase function constructed with 750 frequencies from a Gaussian kernel with $\sigma = 2$ using a bag size of 1000 observations, with some additional Gaussian noise.

Figure F.4: The top and bottom graph denotes the difference in the real and imaginary part of the characteristic function for the $\chi^2(4)/4$ and $\chi^2(8)/8$ as in figure F.2.

Figure F.5: The top and bottom graph denotes the difference in the real and imaginary part of the phase function for the $\chi^2(4)/4$ and $\chi^2(8)/8$ as in figure F.3.

## F.3 Two-Sample Tests with Invariances

### F.3.1 Synthetic $\chi^2$ Dataset

Figure F.6: Extra Type I error results for the synthetic example with $\chi^2$ **Left:** With no noise added for the ME, PhD and SME test. **Right:** Various additive Gaussian components, our base distribution without addition of noise is $\chi^2(4)/4$. Here $n_{11}$ refers to the noise to signal ratio for the first set of samples and $n_{12}$ refers to the second set of samples.

In figure above, the black dashed line is the $99\%$ Wald interval $\alpha \pm 2.57\sqrt{\alpha(1-\alpha)/1000}$, where here $\alpha = 0.05$ is the significance level and $1000$ is the number of repetitions.

On the left figure, we see that indeed all three test considered in this paper indeed controls the Type I error, when the underlying distribution between the two sets of sample is the same, note here no additional noise is added.

On the right figure, we see that the PhD statistic controls Type I error for no added Gaussian noise, and also control Type I error for small differences in additive Gaussian components, unlike the ME test. However, we see that the type I error for a larger noise to signal ratio on the two set of samples indeed does alleviate the Type I error. This is not surprising, as the null distribution was constructed by using a permutation test, using:

$$\varphi_{null} = \frac{1}{2}\varphi_{X_0}\varphi_U + \frac{1}{2}\varphi_{X_0}\varphi_V = \varphi_{X_0}(\frac{1}{2}\varphi_U + \frac{1}{2}\varphi_V),$$

and if the estimated phase features are biased, in the regime with large additive Gaussian noise, then the following may not be true approximately: $\hat{\rho}_{null} = \hat{\rho}_{X_0} = \hat{\rho}_{Y_0}$, leading a to a biased null distribution.

In practice, if it is subtle effects we are looking for, with larger samples, we recommend the use of the SME test, however if this is not the case, then the PhD test is more appropriate, as it has good power for low sample size. In fact, the PhD test has power comparable with that of the ME test, however users should use it with caution, as it does not control the Type I error for larger additional SPD differences and requires more computational power.

### F.3.2 Higgs Dataset

Table F.1: Power for various sample size for high level features of the Higgs dataset

| Sample Size $N$ | SME Power | ME Power |
|---|---|---|
| 500 | 0.94 | 1.0 |
| 600 | 0.969 | 0.999 |
| 700 | 0.987 | 1.0 |
| 800 | 0.989 | 1.0 |
| 900 | 0.994 | 1.0 |
| 1000 | 0.995 | 1.0 |

The table here refers to the high level features of the Higgs dataset, which have been shown to be discriminative in [1]. In this case, clearly both the ME and SME achieve good power, note here the SME has slightly less power, due to using only half of the samples to keep independence.

Figure F.7: Type I error for the Higgs Dataset. **Left:** Extremely low level features **Right:** High level features. The black dashed line is the $99\%$ Wald interval $\alpha \pm 2.57\sqrt{\alpha(1-\alpha)/1000}$, where here $\alpha = 0.05$ is the significance level and $1000$ is the number of repetitions.

The two figures here show that the Type I error is controlled for the ME and SME test, when we have $X_0 \overset{d}{=} Y_0$, where we only consider samples drawn from $Y$, corresponding to the distribution of the processes where the Higgs Boson are produced. Note that on the right graph, the Type I error at first may be slightly alleviated due to small set of samples.

### F.3.3 Aerosol Dataset

We here provide some additional results for the Aerosol Dataset. First, we provide the average RMSE on the aerosol dataset (without noise on test set), based on 10 runs, for different train and test splits in Table F.2.

Figure F.8: Histograms for the distribution of the $L_2$ norm of the averages of Fourier features over each frequency $w$ for the original aerosol test set and the aerosol test set with added noise ($\sigma = 3$), here red line denotes the unit norm representing the phase features **Top Green:** Random Fourier Features $w$ (with the optimised kernel bandwidth) **Bottom Blue:** Learnt Fourier features $w$ from the Fourier Neural Network.

Table F.2: Average RMSE for the Aerosol Dataset across 10 runs, for different train and test splits, with standard deviation in brackets

|          | FOURIER NN     | PHASE NN       | GLRR           | PLRR           |
|----------|----------------|----------------|----------------|----------------|
| NO NOISE | 0.101 (0.011)  | 0.101 (0.008)  | 0.079 (0.010)  | 0.085 (0.009)  |

In the experiments for the Aerosol covariate shift and above, we have seen that the Fourier NN performs similarly to the Phase NN, even under the addition of Gaussian noise, here we provide some possible insights. From the trained Fourier NN on the original dataset, we extract the frequencies $w$ learnt and compute $\left\| \hat{\mathbb{E}} \xi_\omega(X) \right\|$ for each frequency over the original and noisy test set, similarly we do this for the frequencies generated from the Gaussian kernel (with the optimised bandwidth on the original aerosol dataset). We show the empirical distribution of both of these in the figure above, we see that the discriminative frequencies learnt on the training data correspond to the Fourier features which are nearly normalised (i.e. they are close to unit norm like phase features, shown by the red line), this may suggest that the learnt frequencies have captured a notion of invariance to additive SPD components on just the training data. This provides insight into good performance of Fourier NN even under the covariate shift. It also indicates that the original Aerosol data potentially has irrelevant SPD noise components that the Fourier NN has learnt to ignore.

# G  Implementation Details

## G.1  PhD two sample test

For the PhD two sample test for the toy dataset, for each of the 1000 runs, we use a permutation size of 400, with the number of frequencies sampled set at 50. Here the frequencies are sampled using the radial frequency distribution, where $\Sigma$ is chosen to be $\sigma^2 \mathbf{I}$, with $\sigma^2$ being the empirical variance of the two set of samples. The Radial Frequency Distribution is defined as follows:

$$\mathbf{w} = R\Sigma^{-\frac{1}{2}}\psi$$

where $\psi \in \mathbb{R}^n$ is uniformly distributed on the $L_2$ unit sphere $\mathcal{S}_{n-1}$, and $R \in \mathbb{R}_+$ is a radius drawn independently from a folded Gaussian $\mathcal{N}^+(0, 1)$. The radial frequency distribution is useful in high dimensions, as unlike the normal distributions, which 'under samples' the low or middle frequencies,

it is able to sample a broader range of frequencies due to its form. By covering a broader range of frequencies, we may be able to 'better encode' information of the distribution represented by the bags, leading to a feature map that is more informative.

## G.2  Aerosol Dataset

For the network, we use a squared loss function with an additional $L_2$ weight decay for regularisation, with a separate regularisation parameter for the two individual layers. For optimisation, we again use ADAM [8] with fixed learning rate decay and 120 epochs, with a batch size of 10. We perform a 3-fold cross validation, and compute the MSE. We tune the learning rate, regularisation parameters and also number of frequencies for the neural network, here we initialise the first layer with Gaussian distribution with standard deviation = $1/\gamma_0$, where $\gamma_0$ denote the median heuristic.

## G.3  Dark Matter Dataset

For all methods we sample frequencies from the normal distribution (with standard deviation = $1/\gamma_0$, where $\gamma_0$ denote the median heuristic.). After sampling a set of frequencies, we tune the scale of the set of frequencies and also the ridge regularisation parameter using the validation set. In particular we use 75 frequencies on the first and second level of the kernel whenever they are used. Note we use the same set of frequencies (at each individual kernel level) across all the methods in a single run to allow for easier comparison, with potentially different scale tuned on the validation set.