[Reviews · NeurIPS 2017]

Reviewer 1



The authors present an approach relying on a Fourier feature based distance between the phase functions of probability distributions. This type of distance, and the corresponding kernel function follow the concept of the Maximum Mean Discrepancy. The phase function based approach allows to introduce a statistical test to compare distributions corrupted by additive noise. The distribution of the noise is assumed to be unknown except some properties, e.g. positivity of the characteristic function. The phase features are also exploitable in learning from samples clustered into labeled bags. The proposed framework then applied on different data sets to evaluate non-parametric two-sample tests. The approach of the paper is a sounding one. Obviously this technique is created for statistical problems with a special structure, but that kind of structure is hard to handle otherwise with elementary approaches. The model looks like a certain error correcting mechanism on measurements perturbed by special type of noise. I think data sources relating more directly to signal decoding problems, specific examples could be communication between mobiles or between satellites, which can provide even more sounding and practical test environment. The conditions exploited might require additional investigations or relaxation, e.g. indecomposable random variable, since to test them could be even harder than the problem presented by the authors.

Reviewer 2



Summary: The authors proposed phase discrepancy between distributions and phase features for embedding distributions, which are based on the use of phase function that is invariant to additive SPD noise components, aiming respectively at two-sample testing and learning on distributions. The phase discrepancy is similar in mathematical form to MMD but is tailored to incorporate invariance to additive SPD noise. As the phase discrepancy induces a shift-invariant kernel (Proposition 1 and 2), direct application of random Fourier features to such a kernel lead to phase features. Rich experimental results well demonstrate the strength of the proposed methods. Comments: - 1 - Line 120, I think the justification of introducing the subset of indecomposable proba measures for which phase functions are uniquely determined is quite insufficient. For example, how would this lack of identifiability limit the utility of the phase discrepancy? Or does this subset properly include interestingly many distributions for application? - 2 - Since MMD is well studied with the RKHS which contributed deeply to the understanding and well guarantees the utility of MMD, could the authors relate the proposed phase discrepancy to some properties of the RKHS?

Reviewer 3



Overview: In this paper the authors present a method for making nonparametric methods invariant to noise from positive definite distributions by utilizing estimated characteristic functions while disregarding magnitude and only considering the phase. The authors apply the technique to two sample testing and nonparametric regression on measures, ie each "sample" is a collection of samples. The authors demonstrate that the techniques introduced behave as expected by performing tests on synthetic data and demonstrate reasonable performance of the new two sample tests and new feature on real and semi-synthetic datasets. Theory: The fundamental idea behind this paper (disregarding the magnitude of a characteristic function) is not terribly revolutionary, but it is elegant, interesting, and likely to be useful so I think worthy of publication. Though the underlying concept is somewhat technical, the authors manage to keep the exposition clear. The authors explain the concept using population version of estimators and give what seem to be very reasonable finite sample estimators, but it would be nice to see some sort of consistency type result to round things off. Experiments: The synthetic experiment clearly demonstrates that the new features behave as one might expect in for a two sample test, but it is disappointing that the technique seems to greatly reduce the test power compared to current nonparametric techniques. This difference is pretty striking in the >=2000 sample regime. While the technique is not effective for the Higgs Boson two sample test its ineffectiveness is in itself very intriguing, the difference between the two classes seems to be some sort of psd noise phenomena. For the last two experiments, the proposed technique works well on the semi-synthetic dataset (real data with synthetic noise added), but not particularly better, on the final real world data experiment. Ultimately the proposed technique doesn't demonstrate decidedly superior testing or regression performance on any real world dataset, although it does slightly outperform in certain sample regimes. Exposition: The paper is a pleasure to read but there are some errors and problems which I will list at the end of this review. Final Eval.: Overall I think the idea is interesting enough to be worthy of publication. --Corrections By Line-- 3: "rarely" should be "rare" 14: "recently" should be "which have recently been" to sound better 53: To be totally mathematically precise perhaps you should state the E is a psd rv again 115: add "the" after "However" 116: I'm assuming that "characteristic function" was meant to be included after "supported everywhere" 119: Should probably mention a bit more or justify why these "cases appear contrived" 123: The notation P_X and P_Y hasn't been introduced, but perhaps it is obvious enough 123: Here K is a function on a pair of probability measures on line 88 K is a function of two collections of data 134: We need some justification that this is actually an unbiased estimator. A lot of the obvious estimators in the MMD and related techniques literature are not unbiased 208: \chi^2(4)/4 is not technically correct, its the sample that is divided by 4 not the measure 268: I'm not familiar with this dataset. Please mention the space which the labels lie in. I'm assuming it is R. Update: I've bumped it up a point after reading the other reviews and rebuttal. Perhaps I was too harsh to begin with.